# Case Report of a DDX41 Germline Mutation in a Family with Multiple Relatives Suffering from Leukemia

**DOI:** 10.3390/biomedicines12010064

**Published:** 2023-12-27

**Authors:** Jan Nicolai Wagner, Maximilian Al-Bazaz, Anika Forstreuter, Mohammad Ibrahim Hammada, Jurek Hille, Dzhoy Papingi, Carsten Bokemeyer, Walter Fiedler

**Affiliations:** 1Department of Oncology, Hematology and Bone Marrow Transplantation with Division of Pneumology, University Medical Center Hamburg-Eppendorf, 20246 Hamburg, Germany; m.al-bazaz@uke.de (M.A.-B.); a.forstreuter@uke.de (A.F.); m.hammada@uke.de (M.I.H.); j.hille@uke.de (J.H.); c.bokemeyer@uke.de (C.B.); 2Institute of Human Genetics, University Medical Center Hamburg-Eppendorf, 20246 Hamburg, Germany; d.papingi@uke.de

**Keywords:** acute leukemia, DDX41, germline mutation, CPX-351, genetics

## Abstract

Introduction: Previously, it was assumed that genetic influence played a minor role in acute myeloid leukemia (AML). Increasing evidence of germline mutations has emerged, such as *DDX41* germline mutation associated with familial AML. Case presentation: A 64-year-old male patient presented with reduced exercise tolerance and shortness of breath. Following confirmation of AML diagnosis, the patient was enrolled into the AMLSG-30-18 study with a requirement for allogenic stem cell transplantation. The sister was initially selected as a fully HLA-matched donor. However, the family history showed risks for familial AML. Due to the striking family history, further diagnostic steps were initiated to detect a germline mutation. Methods: Using NGS in the patients’ bone marrow AML sample, a *DDX41* mutation with a VAF of 49% was detected, raising the possibility of a germline mutation. DNA from cheek swabs and eyebrows were tested for the presence of the *DDX41* mutation in all siblings. Results: *DDX41* germline mutation was detected in 5 out of 6 siblings. The sister was excluded as a related donor and the search for an unrelated donor was initiated. Conclusion: Obtaining family history of cancer patients plays a crucial role in oncology. If a germline mutation is suspected, further family work-up should be initiated.

## 1. Introduction

Acute myeloid leukemia (AML) is a hematologic malignancy in which there is an excessive proliferation of immature, nonfunctional blasts in the bone marrow [1]. Historically, it was assumed that genetic influence played a minor role in AML development. Increasing evidence of germline mutations has emerged in recent decades. After the discovery of *RUNX1* in 1999 [2], other genes predisposing to myeloid neoplasia were discovered. These include *ANKRD26*, *CEBPA*, *ETV6*, *GATA2* and *DDX41* [3,4]. *DDX41* is a gene encoding an RNA helicase called DEAD-box RNA helicase 41 [5]. The fifth WHO classification of haematolymphoid tumours was revised in 2022 and DDX41-mutated leukemias were classified as genetic tumor syndromes in the subgroup of DNA repair and genomic stability defects [6]. Germline mutation of *DDX41* was first reported in 2015; however, somatic mutation also exists [7]. It has been noticed that the onset of myeloid neoplasm with germline *DDX41* mutation is relatively late, with a median age of 62, and most patients do not show any abnormalities until the onset of the hematological disorder [8]. In recent years, several new studies on the *DDX41* germline mutation have provided new information and insights. However, much is still unclear.

## 2. Detailed Case Description

A 64-year-old man presented with an initial diagnosis of AML. A routine blood examination showed bicytopenia with low hemoglobin and leukocyte levels, which resulted in referral to an oncologist. A bone marrow aspiration was performed, showing more than 20% myeloid blasts, which led to the diagnosis of AML.

The patient reported no symptoms except for a decrease in physical resilience. A review of other systems was negative for fever and infection, as well as for bleeding. The past medical history included arterial hypertension, a previous smoking history of about 20 pack-years and occasional alcohol consumption. At the beginning of the year, he was diagnosed with a deep vein thrombosis (DVT) involving calf, popliteal and femoral vessels. A residual thrombus in the femoral vein was diagnosed in August. The patient had his appendix removed and has total hip joint endoprostheses on both sides.

Professionally, he had worked as a screed installer.

The family history revealed that the patient’s brother and father had hematologic neoplasms. The brother was diagnosed with acute myeloid leukemia in 2017, which progressed from a previously diagnosed myelodysplastic syndrome (MDS). There were no cytogenetic abnormalities. The brother was treated with standard chemotherapy consisting of cytarabine and daunorubicin. After the first induction cycle, 10% blasts were found in the bone marrow, leading to a second induction cycle using high-dose cytarabine and mitoxantrone (HAM protocol). Due to treatment failure on the primary induction, an allogeneic hematopoietic cell transplantation (HCT) from a male 9/10 HLA mismatch was performed, which led to a complete remission since then. Next-generation sequencing (NGS) analysis of a stored DNA sample confirmed the *DDX41* mutation. Another affected brother is currently diseased with a clear-cell renal cell tumor, and the father had suffered from chronic lymphocytic leukemia (CLL); clinical findings in this regard are no longer available.

### 2.1. Clinical Findings

The 64-year-old patient was in a stable general condition and had obese nutritional status. The physical examination was unremarkable except for verrucae vulgares on the back of the right foot.

### 2.2. Diagnostic Assessment

Further work-up was performed according to guidelines. Echocardiography and pulmonary function tests, as well as viral serologies for hepatitis B and C and HIV, were unremarkable. A chest computer tomography scan showed an aneurysmatic dilatation of the aortic bulb (about 4.4 cm) and a subtle ectasia of the aorta ascendens with a diameter of 3.7 cm. There was no evidence of extramedullary involvement of AML.

The cytogenetic analysis revealed a normal male karyotype and the cytomorphology showed a myeloblast percentage of 0% in peripheral blood and 24% in bone marrow, respectively. The bone marrow smear of the patient is shown in Figure 1.

The flow cytometry (FACS) analysis showed positivity for CD34, CD33, CD117 and CD133, which are indicative of AML. The molecular genetic analysis showed a *DDX41* mutation (reference sequence number: NM__016222.2) with a variants allele frequency (VAF) of 49%. A NGS myeloid panel specially adapted to AML genes was used for the test. NGS is a recent technology for targeted DNA sequencing. The molecular analysis using the NGS myeloid panel comprises 20 kb and included the following genes: *ABCA1*, *ASXL1*, *ASXL2*, *BCOR*, *BCORL1*, *BRAF*, *CALR*, *CBL*, *CEBPA*, *CSF3R*, *CSNK1A1*, *DDX41*, *DNMT3A*, *ETNK1*, *ETV6*, *EZH2*, *FLT3*, *GATA2*, *GNB1*, *IDH1*, *IDH2*, *JAK2*, *KDM6A*, *KIT*, *KRAS*, *MPL*, *MYC*, *NF1*, *NPM1*, *NRAS*, *PHF6*, *PPM1D*, *PTPN11*, *RAD21*, *RUNX1*, *SETBP1*, *SF3B1*, *SMC1A*, *SMC3*, *SRSF2*, *STAG1*, *STAG2*, *TET3*, *TP43*, *U2AR1*, *WT1*, *ZBTB7A* and *ZRSR2*. Apart from the *DDX41* mutation, no other mutations were detected. The bioinformatic analysis was performed according to Thol et al. [9]. The sensitivity for the retention of pathogenic variants in the analyzed genes is approximately 95%. The sequencing maturity is at least 50 sequences and a VAF cutoff of \u22655% was applied.

In synopsis of all findings, the patient was classified according to the WHO classification from 2022 regarding myeloid neoplasms with germline predisposition without a pre-existing platelet disorder or organ dysfunction. According to European Leukemia Net (ELN), he fell into the intermediate risk AML group as of 2022 [10]. The hospital’s leukemia board recommended the patient’s enrollment in the AMLSG 30-18 trial. In this study, the intensive standard chemotherapy consisting of daunorubicin and cytarabine is compared with CPX-351 (Vyxeos^®^). The therapy involved two cycles of induction therapy followed by a hematopoietic stem cell transplantation.

### 2.3. Timeline

The patient was enrolled in the AMLSG 30-18 study and randomized to the Vyxeos (CPX 351) arm. The first induction cycle was started in July 2023. A complete hematologic remission was achieved (Figure 2).

The second induction cycle was initiated at the end of September. Complications included anemia (CTCAE III), thrombocytopenia (CTCAE IV) and a catheter-associated infection (CTCAE III). The patient was still in remission in the control. The patient is currently on the stem cell transplant ward.

### 2.4. Molecular Testing

Due to familial clustering, the suspicion of a genetic variant led to the initiation of further diagnostics.. There are no specific guidelines for family screening yet; however, in germline mutations, it is strongly recommended. Testing of frozen bone marrow sample of the brother who suffered from AML revealed a *DDX41* mutation, as well as a *TET2* mutation (Figure 3). The *DDX41* allele frequency ratio was between 38% and 53% and thus compatible with a genetic alteration. Pathogenic variants are assumed to be of germline origin when the allele frequency is approximately 50%, but a range between 30% and 70% is commonly accepted [11]. HLA typing revealed DRB-1 01:01,04:01. As a *DDX41* mutation with 49% VAF was also detected in the patient’s cheek swab, our suspicion of a germline mutation was confirmed.

Initially, one sister was considered as a suitable donor. We arranged for *DDX41* testing of the sister after obtaining consent according to the German Genetic Diagnostics Act. At the same time, a search for an unrelated donor was started. The tests showed that the sister was positive for *DDX41* mutation with a VAF of about 50% (Table 1).

In view of the family findings (see Appendix A), it can be assumed that all three variants are present on the same *DDX41* allele. The two variants detected (c.404G>T [p.(Gly135Val)] and c.1046T>C [p.(Met349Thr)]) have not yet been recorded in the specialist literature or in genetic databases (HG) (MD Professional, ClinVar or in the gnomAD population database).

### 2.5. Follow-Up and Outcomes

Based on the results, the *DDX41*-positive tested sister, who was originally planned as a donor for stem cell transplantation, was then classified as an ineligible donor. A fully matched third-party donor was selected.

Due to the confirmation of the germline mutation an extensive diagnostic workup of the whole family was initiated.

Of the six siblings, five tested positive for the *DDX41* mutation (Figure 3). The *DDX41*-positive tested relatives were admitted to our consultation for hereditary diseases for early disease detection. All family members who tested positive were offered a blood test after receiving a detailed explanation of the advantages and disadvantages of early detection.

Studies have shown that a proportion of *DDX41* mutant carriers had cytopenias or other complete blood count (CBC) abnormalities prior to MDS/AML diagnosis [8]. Other blood count changes described with *DDX41* carriers are macrocytosis and monocytosis as well as antecedent cytopenia [7,12,13]. Consequently, although precise guidelines are not currently available, CBCs can be useful in the screening or surveillance of carriers found by familial screening.

Affected siblings with age over 40 were also offered a bone marrow puncture. We chose the age of 40 as a cut-off value, because of the fact that patients with three significant pathogenic mutations had a minimal risk of myeloid neoplasm before the age of 40, but after that age, the risk increased quickly [14]. The three siblings who underwent bone marrow puncture showed unremarkable results.

## 3. Discussion

Contrary to historical opinion, pathogenic or likely pathogenic germline mutations play a role in approximately 14% of patients suffering from acute myeloid leukemia [4]. These genes include, among others, *ANKRD26*, *CEBPA*, *ETV6*, *GATA2* and *DDX41* [3,15,16,17], while *DDX41* is one of the most frequent ones [4]. It is located at 5g35.3 and the mutation is thought to be present in 1.5–6% of AML patients [12,18] and is inherited in an autosomal-dominant manner. This mutation is reported to increase the risk of myeloid neoplasm, with a 3:1 male predominance, suggesting a gender-specific effect on myeloid leukemogenesis and a long-term cytopenia before development of myeloid neoplasm. In AML, *DDX41* was identified as a tumor suppression gene, as overexpression blocks cell proliferation while knockdown promotes tumor growth [7]. Insufficient *DDX41* also promotes the accumulation of R-loops, which leads to the proliferation of hematopoietic cells through inflammatory signals [19]. The R-loop itself is a three stranded nucleic acid structure composed of an RNA/DNA hybrid and a displaced single-stranded DNA. It plays versatile roles in many physiological and pathological processes. A total of 47% of patients show other somatic mutations, such as *ASXL1*, *TP53* and *TET2*, as seen in the brother, who suffered from AML [20]. An exact risk is not yet known. As seen in our patient, the bone marrow testing usually shows hypocellularity and a normal karyotype [20]. Interestingly, CD13 was not expressed on the patients blasts. It could be further investigated whether non-expression of CD13 is common in DDX41 mutations or familial leukemias. To the best of our knowledge, this has not yet been investigated in any studies.

### 3.1. New DDX41 Variant

The *DDX41* variant c.1372G>T leads to the appearance of a premature stop codon in the mRNA [p.(Glu458*)]. It is to be expected that this change will result in a so-called functional null allele. In the case of a null allele, the gene product is no longer produced correctly or is not functional. As the *DDX41* variant c.1372G>T [p.(Glu458*)] is associated with an expected loss of protein function, the variant is assessed as probably disease-relevant (IARC class 4) according to the ACMG evaluation criteria (criteria applied: PVS1, PM2).

The *DDX41* variants c.404G>T [p.(Gly135Val)] and c.1046T>C [p.(Met349Thr)] have also not yet been recorded in the literature or in genetic databases. In silico analyses for pathogenicity assessment either rate these two variants as neutral or show no clear tendency regarding a possible pathogenetic relevance. According to the ACMG evaluation criteria, the variants c.404G>T [p.(Gly135Val)] and c.1046T>C [p.(Met349Thr)] are evaluated as variants of unclear clinical significance (IARC class 3; criteria applied: PM2, BP2 or PM1, PM2, PM5, BP2) based on current data. Since the variants c.404G>T [p.(Gly135Val)] and c.1046T>C [p.(Met349Thr)] were co-inherited, it can be assumed that they are located on the same chromosome.

The risk for lymphoid neoplasm is also increased including non-Hodgkin lymphoma, Hodgkin lymphoma, multiple myeloma, monoclonal gammopathy of undetermined significance, chronic lymphocytic leukemia (as seen in the father) and acute lymphoblastic leukemia [8,12,13,21]. Furthermore, *DDX41* expression is associated with the occurrence of clear-cell renal cell carcinomas, as detected in the brother [22].

### 3.2. Treatment Options

At this time, there is no guidance for the treatment of *DDX41* germline mutant acute leukemias; however, patients should undergo early allogeneic stem cell transplantation in addition to standard chemotherapy [23,24]. However, in *DDX41*-mutated MDS patients, good response rates of 70% have also been shown under therapy with hypomethylating agents (HMA) [25]. In AML, one study reported a response rate of 78% for treatment with HMA with or without venetoclax [26]. However, a higher response rate of 94% was achieved with intensive chemotherapy in intermediate or adverse risk patients [27].

Stem cell transplant from related donors who have tested positive for *DDX41* should be avoided [28,29]. However, there is no absolute contraindication due to the lack of data.

Several germline *DDX41* variants are prevalent among specific populations. Kovilakam et al. found that one in 129 individuals in the UK biobank had a *DDX41* variant [30]. Many allogeneic donors—related or unrelated—have a germline *DDX41* variant, which can therefore be passed to the recipient during allogeneic hematopoietic stem cell transplantation [29,31]. Therefore, Makishima et al. advocate to add *DDX41* to all molecular profiling panels and testing for germline *DDX41* in all allogenic stem cell donors [32]. It should also be noted that stem cell grafts from *DDX41* mutant patients are more likely to induce severe graft-versus-host disease (GvHD) (stage 3 to 4) when transplanted into wild-type carriers [33]. It is postulated that patients with germline *DDX41* variants have a proinflammatory milieu in all of their nontransplant organs that is created by the germline allele, which activates donor-derived T cells and results in severe GvHD [32]. Post-transplant application of cyclophosphamide showed promising results in preventing GvHD [33].

Some studies also reported responsiveness to lenalidomide for *DDX41*-associated familial MDS/AML [7,34,35]. Overall, patients with *DDX41*-related MDS/AML have a relatively favorable outcome with 2-year survival rates of 90%, not reaching the median OS in a follow-up of 2.8 years [12]. In a propensity score-matched study with *DDX41* mutations, it could be demonstrated that the median overall survival (OS) of germline-mutated patients was longer (5.2 years) than that of the patients with wild-type *DDX41* (2.7 years); however, this difference was not statistically significant [12]. In an age-matched study comparing AML cases with *DDX41* or a variant of unknown significance, the superior OS could also be shown [36].

### 3.3. Risk to Family Members

It is critical to screen the patient’s parents for the germline mutation, as the result is crucial to assess the risk of the siblings. In our patient, unfortunately, both parents are already deceased, so we could not perform testing. However, due to the diagnosis of CLL in the father, we assume positivity. Then, the risk of the siblings of inheriting the pathogenetic variant is 50% due to autosomal transmission, whereas if no DDX41 mutation in the parental leukocyte DNA can be detected, then the risk is 1% due to the theoretic possibility of parental germline mosaicism [37].

For *DDX41* mutant family members, it is recommended to consider testing of symptomatic individuals regardless of age and to perform differential blood counts every 6–12 months [23]. In addition, a history of clinical symptoms of MDS/leukemia should be obtained and patients should be clinically examined annually. There is no clear recommendation for a routine bone marrow aspiration [23]; however, we offered bone marrow aspiration to all first-degree relatives. Furthermore, genetic counseling is advised.

## 4. Conclusions

We have identified a family in which a previously undescribed *DDX41* germline mutation is present. The importance of a detailed family history and testing is emphasized here, especially when a stem cell transplant is planned.

Several oncological diseases have been described in the family history and we have included the relatives in a screening program.

Early detection of AML predisposition is of a great importance in ensuring the best possible early diagnosis and therapy. Monitoring relatives to detect blood count changes early should be offered. It is to be hoped that molecular early detection will be made possible in the future.

In addition, it is crucial to identify stem cell transplant donors early in the family to prevent donor-derived leukemia from germline carriers. When patients are transplanted, particular attention should be paid to the increased risk of GvHD and appropriate therapies should be initiated at an early stage.

Further research results are needed to identify additional germline mutations in the future and to put the new knowledge into context.

## 5. Patient Perspective

Except for the shortness of breath, I never really had any health problems. Lately, climbing stairs has been a little more difficult for me. When my blood was tested by my family doctor and abnormalities were discovered, which required further work-up, I immediately thought that this could be leukemia. Since my father and my brother both had the disease, I thought it might be my turn now. Because I had experienced the disease from my brother having a good outcome, I felt safe to undergo the therapy.

## Figures and Tables

**Figure 1 biomedicines-12-00064-f001:**
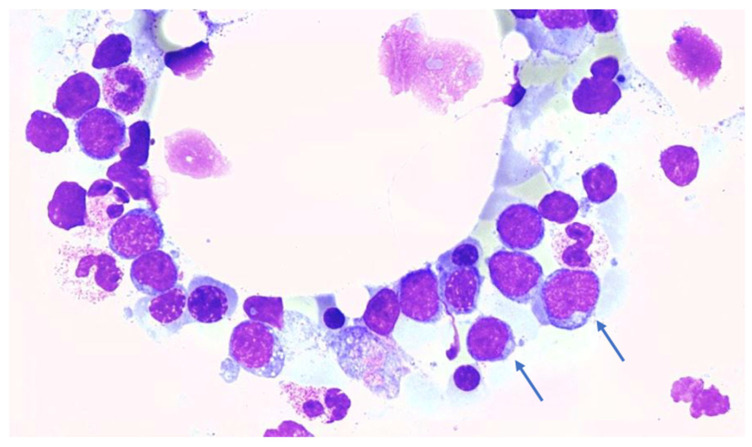
Patient blood smear with several blasts. A hypercellular bone marrow with many blasts (arrows) is visible. 30× magnification. Blasts present as medium-sized cells with a round-oval nucleus and a narrow deep basophilic cytoplasmic rim. Neutrophil granulocytes are also visible. The bone marrow shows a pathologically characteristic proliferation of blasts of 24% (less than 5% in normal bone marrow). Source: University Medical Center Hamburg-Eppendorf; Oncology Center; II Medical Clinic, Cytomorphology.

**Figure 2 biomedicines-12-00064-f002:**
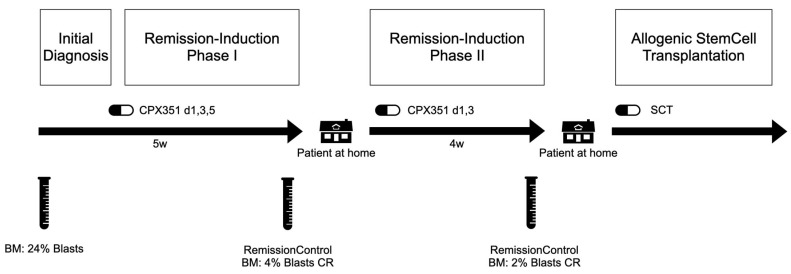
Timeline of planned oncological procedure.

**Figure 3 biomedicines-12-00064-f003:**
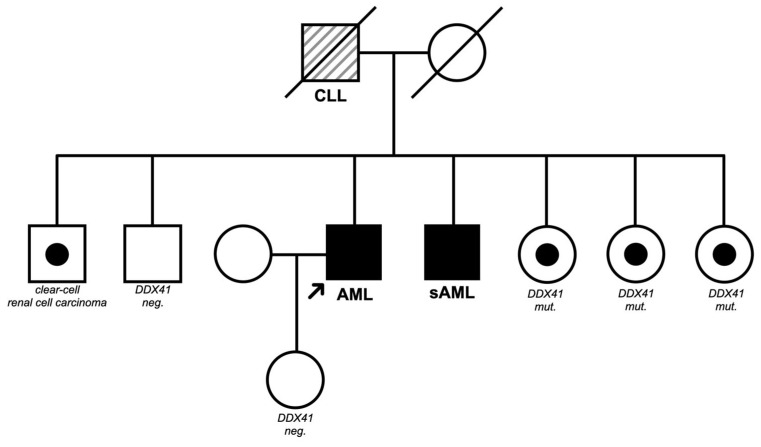
Family tree of the patient.

**Table 1 biomedicines-12-00064-t001:** *DDX41* analyses by NGS method. VAF; variants allele frequency, % mutated by NGS. Reference sequence number: NM__016222.2.

Person	DNA/cDNA	Protein	VAF (%)
Patient	c.404G>T	p.Gly135Val	49
	c.1046T>C	p.Met349Thr	47
	c.1372G>T	p.Glu458Ter	48
Brother 1	c.404G>T	p.Gly135Val	53
	c.1046T>C	p.Met349Thr	52
	c.1372G>T	p.Glu458Ter	38
Sister 1	c.404G>T	p.Gly135Val	52
	c.1046T>C	p.Met349Thr	48
	c.1372G>T	p.Glu458Ter	50

## Data Availability

The data presented in this study are available on request from the corresponding author. The data are not publicly available due to patients’ privacy.

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
