# Peer review of "Case Report of a DDX41 Germline Mutation in a Family with Multiple Relatives Suffering from Leukemia"

_biomedicines, 2023, doi:10.3390/biomedicines12010064_

Round 1

Reviewer 1 Report

Comments and Suggestions for Authors

This is a case report of a male adult with acute myeloid leukemia (AML). This disease is the most frequent leukemia in the adults, representing around 80% of the cases. It is characterized by a clonal expansion of immature blast cells in the peripheral blood and bone marrow. As a result, there is innefective erythropoiesis and bone marrow failure. The prognosis is unfavorable above 60 years.

(1) Line 28 states: "A 64-year-old male patient presented with reduced exercise tolerance and shortness of breath". However, line 91 states "54-year-old patient in stable general condition and obese nutritional status.".

Could you please confirm if it is 64 or 54?

(2) Depending upon the etiology, genetics, immune-phenotype, and morphology, there are different classification systems for AML. It is important to describe the classification of AML properly in the Introduction of the manuscript to help the readers not familiar with this topic.

(2.1) FAB classification from M0 to M7

(2.2) WHO revised 4th edition of 2016 (AML with recurrent genetic abnormalities, AML with myelodysplasia-related changes, therapy-related myeloid neoplasms, NOS, myeloid sarcoma, myeloid proliferations related to Down syndrome)

2.2.1.AML with recurrent genetic abnormalities

2.2.2. AML NOS

2.2.3. AML de novo or seondary (prior MPD or MDS, etc.)

(2.3) Arber DA, Orazi A, Hasserjian RP, et al. International Consensus Classification of Myeloid Neoplasms and Acute Leukemias: integrating morphologic, clinical, and genomic data. Blood. 2022;140(11):1200-1228. doi:10.1182/blood.2022015850

***Table 24.
ICC of hematologic neoplasms with germline predisposition
Hematologic neoplasms with germline predisposition without a constitutional disorder affecting multiple organ systems
 Myeloid or lymphoid neoplasms with germline DDX41 mutation

(2.4) Khoury, J.D., Solary, E., Abla, O. et al. The 5th edition of the World Health Organization Classification of Haematolymphoid Tumours: Myeloid and Histiocytic/Dendritic Neoplasms. Leukemia 36, 1703–1719 (2022). https://doi.org/10.1038/s41375-022-01613-1

*** In this article it is written that "Individuals with germline pathogenic variants in GATA2, DDX41, Fanconi anaemia (FA) or telomerase complex genes can have hypoplastic bone marrow and evolve to MDS and/or AML and do not respond to immunosuppressive treatment" in the section "MDS, morphologically defined".

***In the section Genetic tumor syndromes with predisposition to myeloid neoplasia, DDX41 is not mentioned. Could you please explain if you are referring to the "beta WHO-5ed" version are is available online upon payment?

(3) Line 56 "These include ANKRD26, CEBPA, ETV6, GATA2, and DDX41". Could you please confirm these genes?

(4) Line 81 "The brother had a secondary acute myeloid leukemia progressing from myelodysplastic syndrome". Could you please provide more details (if available)?

(5) Line 68 "During a routine blood examination, abnormal blood values were noticed, which resulted in referral to an oncologist. A bone marrow aspiration was performed, which led to the diagnosis of AML". Please provide more exact details of the diagnosis.

(6) Line 103 "The cytogenetic analysis revealed a normal male karyotype and the cytomorphology 103 showed a myeloblast percentage of 0% in peripheral blood and 24% in bone marrow". Please confirm this statement. 0% in periphelral blood? (The presence of at least 20% blasts in the bone marrow or peripheral blood is diagnostic of AML).

(7) Please confirm the flow cytometry markers.

(8) Regarding the DDX41 mutational analysis.

https://www.illumina.com/products/by-type/clinical-research-products/trusight-myeloid.html#gene-list

I am sorry as it may be my mistake, but I cannot identify the DDX41 in the list of genes of the array.

Please confirm.

(9) Line 113. Please define ELN 2002 risks.

(10) Line 114. Please define and describe the AMLSG 30-18 clinical trial.

(11) Figure 1. Please describe the cytological image. Please explain why it is characteristics of AML.

(12) Please provide the complete annotation file for the mutations of DDX41 and the other genes. You may show the ANNOVAR file and the filtered one with the most relevant / pathogenic mutations.  Please describe the NGS procedure with more details.

(13) Is the patient perspective section relevant?

Author Response

We want to thank the reviewers for their helpful comments which help to make the manuscript clearer and more precise.

1)Thank you for pointing this out, we have corrected the typo, 64 years is the correct age.

2) According to the newest WHO classification 2022 the patient falls into the category of Myeloid neoplasms with germline predisposition without a preexisting platelet disorder or organ dysfunction

3) We confirm that these genes play a role in germline mutation, as shown in the cited papers.

4) As requested, we have included further details in the report.

5) Done as requested

6) We confirm this statement. With a blast percentage of 24% in bone marrow the diagnosis of AML is confirmed.

7) We confirm

8) Thank you for your inquiry. We have explained it in more detail in the text, the Ilumina DNA Prep was used, with an IDT enrichment. We used a "custom myeloid panel" for targeted sequencing, which we have adapted slightly to the "normal myeloid panel" from Illumina according to the AML genes. DDX41 is therefore included in our panel and is therefore sequenced for everyone. We have asked if we can get the Anovar files, the answer is still pending.

9) Done as requested

10) Done as requested

11) Done as requested

12) Done as requested, Anovar file pendiding

13)We have written the case report according to the CARE case report guidelines, which requires the perspective section.

Reviewer 2 Report

Comments and Suggestions for Authors

The manuscript describes "Case report of a DDX41 germline mutation in a family with multiple relatives suffering from leukemia." This article shows that genetic influences, such as DDX41 germline mutations, play an important role in acute myeloid leukemia (AML). This gene plays a crucial role in oncology, but several points need clarification.

Comment:

1. Figure 1. A control group should be added to the blood smear

2. Table 1. DDX41 analysis by NGS method. DNA and protein data for controls should be added. The author should explain it in the text to make it easier for readers to read.

Comments on the Quality of English Language

 Minor editing of English language required

Author Response

We want to thank the reviewers for their helpful comments which help to make the manuscript clearer and more precise.

1.: The control group would be healthy individuals. We prefer not to show normal blood smears because this adds little to the conclusion.

2.: The NGS panel is performed with DNA. As requested, we have explained the NGS method in more detail in the text. NGS analysis of normal individuals would not add in this regard. Unfortunately, we do not have a protein sample to perform Western blots. But we think that mutational analysis is sufficient to prove the genetic detect.

Reviewer 3 Report

Comments and Suggestions for Authors

It is believed that AML in DDX4 germline variant carriers is usually associated with a second hit inactivating DDX4 allele.

Surprisingly, the authors found no other variants whatsoever in myeloid chip. For a sound analysis the authors shoould also run genomic microarrays to exluded copy number loss at that locus.

Would it make sense to monitor CHIP in the carriers in the family?

Author Response

We want to thank the reviewers for their helpful comments which help to make the manuscript clearer and more precise.

Since the patient had a normal karyotype, major gains or losses in DNA seem unlikely. Unfortunately, no further DNA sample of the patient is available for a microarray analysis.

To monitor CHIP in affected family members seems appealing, but a literature search did not result in any hit for the presence of CHIP in DDX41 carriers therefore, we preferred not to do this analysis.

Round 2

Reviewer 1 Report

Comments and Suggestions for Authors

Thank you for the answers.

Please upload the fastq files or the annotation files. Since many genes were analyzed and only was found to be mutated, I wonder if lower confidence mutations were indeed found.

Reviewer 3 Report

Comments and Suggestions for Authors

The explanations of the authors are satisfactory, the paper seems worth publishing.

Comments on the Quality of English Language

No problems with English